# Evaluation of Catechin Synergistic and Antibacterial Efficacy on Biofilm Formation and *acrA* Gene Expression of Uropathogenic *E. coli* Clinical Isolates

**DOI:** 10.3390/antibiotics11091223

**Published:** 2022-09-09

**Authors:** Najwan Jubair, Mogana R., Ayesha Fatima, Yasir K. Mahdi, Nor Hayati Abdullah

**Affiliations:** 1Faculty of Pharmaceutical Sciences, UCSI University, Kuala Lumpur 56000, Malaysia; 2Beykoz Institute of Life Sciences and Biotechnology, Bezmialem Vakif University, 34820 Istanbul, Turkey; 3Forest Research Institute Malaysia (FRIM), Kuala Lumpur 52109, Malaysia

**Keywords:** *Canarium patentinervium* Miq., multidrug resistance, antibiofilm activity, AcrAB-TolC efflux pump, molecular docking, Autodock Vina

## Abstract

Uropathogenic *Escherichia coli* has a propensity to build biofilms to resist host defense and antimicrobials. Recurrent urinary tract infection (UTI) caused by multidrug-resistant, biofilm-forming *E. coli* is a significant public health problem. Consequently, searching for alternative medications has become essential. This study was undertaken to investigate the antibacterial, synergistic, and antibiofilm activities of catechin isolated from *Canarium patentinervium* Miq. against three *E. coli* ATCC reference strains (ATCC 25922, ATCC 8739, and ATCC 43895) and fifteen clinical isolates collected from UTI patients in Baghdad, Iraq. In addition, the expression of the biofilm-related gene, *acrA*, was evaluated with and without catechin treatment. Molecular docking was performed to evaluate the binding mode between catechin and the target protein using Autodock Vina 1.2.0 software. Catechin demonstrated significant bactericidal activity with a minimum inhibitory concentration (MIC) range of 1–2 mg/mL and a minimum bactericidal concentration (MBC) range of 2–4 mg/mL and strong synergy when combined with tetracycline at the MBC value. In addition, catechin substantially reduced *E. coli* biofilm by downregulating the *acrA* gene with a reduction percent ≥ 60%. In silico analysis revealed that catechin bound with high affinity (∆G = −8.2 kcal/mol) to AcrB protein (PDB-ID: 5ENT), one of the key AcrAB-TolC efflux pump proteins suggesting that catechin might inhibit the *acrA* gene indirectly by docking at the active site of AcrB protein.

## 1. Introduction

*Escherichia coli* is a Gram-negative multifaceted bacterium that comprises commensal *E. coli*, which normally colonizes in the gastrointestinal tract of humans and animals a few hours after birth [1], and pathogenic *E. coli*, which is the most common cause of gastrointestinal infections such as diarrheal disease caused by enterotoxigenic *E. coli* (ETEC) and extraintestinal infections such as urinary tract infections (UTI) caused by uropathogenic *E. coli* (UPEC) [2]. Some *E. coli* clones became pathogenic when they gain a number of specialized virulence factors, such as different adhesins, toxins, siderophores, and iron acquisition systems that interfere with the way the host cell works [3].

UTIs are commonly classified as community-acquired or healthcare-associated UTIs, and as complicated or uncomplicated UTIs, depending on the severity of the infection [4]. UTI is one of the most common bacterial infections worldwide, with an estimated 150 million UTI cases occurring each year [5]. Due to the anatomy of the female urinary tract, it is estimated that 50–60% of women will develop an UTI at some point in their lives [6]. UTIs are treatable in most cases. However, the progression of multidrug-resistant strains results in recurrent infections, treatment failure, and complications associated with increased rates of mortality and morbidity [7].

*E. coli* resists antibiotics through several mechanisms, such as production of enzymes called “beta-lactamases”, which is a group of more than 2800 compounds derived from environmental sources, including extended spectrum beta-lactamase (ESBL), AmpC beta-lactamase that acquires resistance to penicillin and cephalosporins, New Delhi metallo-beta-lactamase, and Carbapenem hydrolyzing oxacillinase-48 [8]. Furthermore, efflux pump activity such as the resistance-nodulation-division (RND) tripartite efflux pump, AcrAB-TolC, represents the major contributor to intrinsic multidrug resistance in *E. coli* [9]. In addition, *E. coli* possesses several virulence factors to ensure survival, such as hemolysis and biofilm formation [10].

Biofilm formation enables *E. coli* colonies to evade the immune system and antibiotics, making it difficult to eradicate and resulting in multiple antibiotic resistance [11]. Increased drug resistance and the emergence of bacterial infections with no treatment options yet available make research into new antimicrobial agents urgent. On the other hand, medicinal plants provide a promising source for drug discovery. Since ancient times, Indigenous peoples have utilized medicinal plants to treat various conditions [12]. Today, people continue to depend on herbal therapy, particularly in developing nations where 80% of the population uses traditional medicine to treat a variety of illnesses [13].

*Canarium patentinervium* Miq. is a rare tropical plant belonging to the family Burseraceae, genus *Canarium* L., native to Asiatic–Pacific region, Malaysia, and Brunei [14]. It has been used for wound healing by Indigenous peoples of Malaysia [14]. Our team previously reported the antibacterial activity of the crude extracts, fractions, and isolated compounds from the leaves and bark of this plant [15,16]. The current study aimed to comprehensively evaluate the antibacterial activity of catechin (Figure 1), one of the active compounds isolated previously from the leaves of *Canarium patentinervium* Miq., against biofilm-forming, uropathogenic *E. coli* isolates from UTI urine samples collected in Baghdad, Iraq during the period November–December 2021. Molecular docking was carried out to achieve a deep insight into the molecular mechanism and to analyze the binding mode between catechin and the target protein proposed in the study.

## 2. Results

### 2.1. Evaluation of the Antibacterial Activity

The study investigated the antibacterial activity of catechin isolated from the leaves of *Canarium patentinervium* Miq. against three *E. coli* ATCC strains (*E. coli* ATCC 25922, *E. coli* ATCC 8739, and *E. coli* 43895) and 15 multidrug-resistant *E. coli* clinical isolates. The results are displayed in Table 1. Among the tested antibiotics, tetracycline, erythromycin, clindamycin, and vancomycin showed insignificant inhibition zones in all *E. coli* ATCC strains. *E. coli* ATCC 8739 had intermediate sensitivity to rifampin (17.24 mm) and gentamicin (13.14 mm). In addition, gentamicin displayed moderate inhibition against *E. coli* ATCC 43895 (13.6 mm), although it showed a significant zone of inhibition against *E. coli* ATCC 25922 (31.32 mm). Nevertheless, *E. coli* clinical isolates displayed insignificant inhibition zones for most of the tested antibiotics. In regard to catechin, *E. coli* ATCC 8739 was resistant (9 mm), although catechin displayed remarkable inhibition against the remaining *E. coli* strains ATCC 25922 (10.47 mm), *E. coli* ATCC 43895 (11.8 mm) and most of the clinical isolates (zone of inhibition ranges from 9.93 mm to 13.92 mm). The synergistic effect of catechin in combination with the tested antibiotics was evaluated in this study. Results are displayed in Table 1. The strongest synergistic effect was observed between catechin and tetracycline, through which all the isolates (*n* = 15) showed synergism with no antagonism reported. In addition, combinations between catechin with azithromycin, gentamicin, erythromycin, and clindamycin displayed a high percentage of synergism with additive effects reported in three isolates, while the combination of catechin with rifampin had the lowest synergy with high antagonism reported.

The minimum inhibitory concentration is the lowest antimicrobial concentration that inhibits the visible growth of a bacterium following overnight incubation [17]. Azithromycin showed potent antibacterial activity with a minimum inhibitory concentration (MIC) of 0.5 mg/mL against *E. coli* ATCC 8739 and 0.5–1 mg/mL against *E. coli* clinical isolates (Table 2). The MICs for the rest of the antibiotics ranged from 2 to 32 mg/mL, and they were only moderately effective against all of the strains tested. Catechin exhibited moderate activity (MIC = 1 mg/mL) against *E. coli* ATCC 25922, *E. coli* ATCC 43895, and *E. coli* clinical isolates, weak activity against *E. coli* ATCC 8739 (MIC = 2 mg/mL), and potent activity against two clinical isolates (isolate numbers 9 and 10) with a MIC of 0.5 mg/mL. According to the literature, strong antibacterial activity is considered with MIC values ranging from 0.05–0.5 mg/mL, moderate activity with MIC values ranging from 0.6–1.5 mg/mL, and weak antibacterial activity when the MIC values exceed 1.5 mg/mL [18].

The minimum bactericidal concentration (MBC) is the lowest antimicrobial concentration that will stop an organism from growing after being subcultured on an antibiotic-free medium [17]. According to the ratio of MBC/MIC, the antibacterial effect is considered bactericidal if the MBC/MIC ≤ 4, and the effect is considered bacteriostatic if the MBC/MIC > 4 [19]. The MBC, MIC, and MBC/MIC ratio for the tested antibiotics and catechin are displayed in Table 2. All the tested antibiotics showed bactericidal activity against the tested strains except for tetracycline, erythromycin, and clindamycin, which showed bacteriostatic activity against some of the clinical isolates. Catechin exhibited bactericidal action against all the tested *E. coli* strains. Azithromycin was the most active antibiotic and *E. coli* ATCC 8739 was the most susceptible strain (MIC = 0.5 mg/mL, MBC = 1 mg/mL, MBC/MIC = 2). It is noteworthy that *E. coli* ATCC 8739 was resistant to catechin and all antibiotics except azithromycin. Interestingly, *E. coli* ATCC 43895 was the most sensitive strain to catechin (MIC = 1 mg/mL, MBC = 2 mg/ mL, MBC/MIC = 2), although it was resistant to all the tested antibiotics. Both catechin and azithromycin had significant antibacterial activity against *E. coli* clinical isolates with bactericidal effects, while the remaining antibiotics were inactive against the tested strains (MIC ranges of 2–32 mg/mL, MBC ranges of 4–64 mg/mL).

### 2.2. Biofilm Inhibition

The results for the biofilm inhibition assay of this study are shown in Figure 2. All the tested *E. coli* strains were biofilm producers. *E. coli* ATCC 43894 and the clinical isolates were strong biofilm formers (mean OD = 0.25), while *E. coli* ATCC 8739 and *E. coli* ATCC 25922 were moderate biofilm formers (mean OD = 0.18). According to the literature, mean OD values > 0.24 indicate that the tested bacterium can form strong biofilm, mean OD values 0.12–0.24 indicate that the test bacterium has moderate biofilm-forming ability, and mean OD values < 0.12 indicate weak or no biofilm formation [20]. Catechin at the MIC level distorted the biofilm formation of all the tested *E. coli*. The results indicated that catechin inhibited *E. coli* adhesion at a percentage of inhibition equal to 90.3% for *E. coli* ATCC 25922, 60% for *E. coli* ATCC 8739, 100% for *E. coli* ATCC 43895, and an average of 82% biofilm inhibition for *E. coli* clinical isolates. Inhibition of biofilm formation by gentamicin was modest against all of the examined *E. coli* strains.

### 2.3. In Silico Study

In a previous study, the relationship of the *acrA* gene with the biofilm formation ability of *E. coli* was established [11]. Thus, a molecular docking study was carried out to check catechin binding affinity to this protein. The AcrA protein is one of the AcrAB-TolC multidrug-resistant efflux pump proteins (Figure 3). AcrAB-TolC is a member of the resistance nodulation division family (RND) that is present in Gram-negative bacteria and has a crucial role in the *E. coli* resistance mechanism to broad spectrum antibiotics [21]. The pump consists of three major proteins; an outer membrane channel called TolC acts as an exit pathway of substrates; periplasmic protein AcrA is responsible for the stability of the connection between AcrB and TolC; and an inner membrane protein called AcrB, which is the target site for substrate binding [22,23,24,25]. In addition to the small residue that was identified recently, AcrZ affects substrate preference of AcrB [26].

It is noteworthy that the AcrA protein has no substrate binding site and AcrB is the substrate binding site in several antimicrobial agents such as Puromycin (PDB-ID: 5NC5), Erythromycin (PDB-ID: 3AOC), Rifampicin (PDB-ID: 3AOD), Levofloxacin (PDB-ID: 7B8T), Doxycycline (PDB-ID: 7B8R), Linezolid (PDB-ID: 4K7Q), Fusidic acid (PDB-ID: 6Q4P), and Minocycline (PDB-ID: 5ENT). In this regard, catechin binding affinity to AcrB was tested and the result was presented in Figure 2. Catechin showed high binding affinity to AcrB (PDB-ID: 5ENT) with ∆G = −8.2 Kcal/mol compared to the control ligand (minocycline) with ∆G = −8.8 Kcal/mol (Table 3). Nevertheless, further in vitro tests on the catechin effect on the AcrAB-TolC are required to confirm efflux pump inhibition of catechin.

### 2.4. Gene Expression

In the current study, fifteen clinical *E. coli* isolates were tested for the expression of the biofilm related gene, *acrA*, using the SYBR green assay for real-time PCR [27]. The results are summarized in Figure 4 and Table 4. A housekeeping gene, 16S rRNA, was used, which is commonly used in PCR analysis of bacteria [28]. The *acrA* gene was expressed in nine *E. coli* isolates (60% target gene expression). However, the expression was depressed in *E. coli* isolates treated with catechin at sub-MIC (0.5 mg/mL).

Ct (thermal cycle) is the number of cycles required for the fluorescent signal to cross the thresholds and it is inversely proportional to the amount of the target gene (i.e., lower Ct indicates greater amount of target gene) [29]. Cts values of 29 indicate an abundance of target genes and strong positive reactions, according to the literature. Cts values ranging from 30–37 indicate a moderate number of target genes. Cts values of 38–40 are considered weak reactions with a small number of target genes [30,31]. In this study, *acrA* expression occurred in Ct < 29, reflecting a positive reaction. Moreover *E. coli* isolates that were treated with catechin showed a significant reduction in the target gene (*acrA*). By combining the results of the in silico and the in vitro studies, we suggested that catechin antibacterial action against multidrug-resistant uropathogenic *E. coli* is due to biofilm inhibition and efflux pump mechanism. According to the in silico result, catechin binds the substrate recognition site in the AcrAB-TolC of *E. coli*, which might indirectly affect the AcrA protein. Further in vitro tests on the catechin activity on the *E. coli* efflux pump system are warranted.

## 3. Discussion

Urinary tract infections (UTIs) are one of the most prevalent infectious diseases with a high recurrence rate worldwide [32]. Among several pathogens that cause UTIs, *E. coli* is considered as the major causative pathogen, accounting for 90% of community-acquired infections and 50% of nosocomial infections [33]. Recurrent UTIs with multidrug-resistant *E.coli* impose both an economic and health burden [34]. It has been reported that 78% of recurrent urinary tract infections are caused by multidrug-resistant, biofilm-forming *E. coli* [33]. Biofilm formation by many pathogens is considered as one of the indirect strategies for multidrug resistance [35]. Bacteria producing biofilm are difficult to eradicate and are able to transfer their resistance genes within their biofilm community, causing recurrent infections [36]. Many species have the ability to form biofilms, such as *Escherichia*, *Pseudomonas*, *Staphylococcus*, *Bacillus*, etc. [37].

A biofilm is a complex matrix composed of polysaccharides, nucleic acids, proteins, lipids, water, various ions, and other organic components in which cells bind together to survive harsh conditions such as host defense and accumulation of various noxious substances and antimicrobial agents [38]. According to the recent assessment by the National Institute of Health (NIH), more than 60% of in vivo infections are due to biofilm-forming microorganisms [39]. Currently, biofilm-producing bacteria have been designated as a major concern because of chronic infections [40]. It has been proven that biofilm formation makes bacteria 10–1000 times more antibiotic resistant. Several pathogens such as *Enterococcus faecium*, *Staphylococcus aureus*, *Klebsiella pneumoniae*, *Acinetobacter baumannii*, *Pseudomonas aeruginosa*, and *Enterobacter* spp. that are known as “ESKAPE” infections are distinguished by their strong biofilm formation [38]. Thus, the search for new antimicrobials with biofilm inhibition properties is urgently needed.

Plant secondary metabolites have acquired extra attention in the area of drug development as they are safe, derived from natural sources, and have high bioavailability [41]. Many of them have been proven to have antibiofilm activity [42]. *Canarium patentinervium* Miq. is a rare plant from the Burseraceae family, genus *Canarium* L., found in Asia [43]. Our team previously documented several medicinal activities of this plant, such as antibacterial, antioxidant, anti-inflammatory, anticholinesterase, and antitumor [16,44,45]. In this study, the antibacterial activity of catechin isolated from the ethanolic extract of the leaves of *Canarium patentinervium* Miq. was assessed against reference and multidrug-resistant uropathogenic *E. coli*. Based on the findings, catechin had significant antibacterial activity (MIC ranges of 1–2 mg/mL, MBC/MIC ≤ 4) against all the tested strains. In addition, catechin showed high synergism with the tetracycline combination. Catechin’s antibacterial effect against *E. coli* has been reported in several studies [46,47,48]. It was shown that catechin inhibited *E. coli* growth in a dose-dependent manner [48,49].

In this study, the biofilm formation ability of *E. coli* strains was assessed quantitatively, and the result indicated that all the tested *E. coli* strains formed biofilm and were multidrug resistant. Catechin inhibited *E. coli* biofilm significantly, with a percent inhibition range of 60–100%. The antibiofilm activity of catechin has been reported in several studies [50,51]. Catechin at a concentration of 0.026 g/L showed significant biofilm inhibition in *MRSA* strains through downregulation of *fnbA* and *icaBC* genes in *MRSA* [50]. Similarly, green tea epigallocatechin gallate EGCG at sub-inhibitory concentration remarkably reduced the adhesion of *MRSA* by interfering with bacterial glucosyltransferase involved in biofilm formation [52]. In *E. coli*, EGCG showed antibiofilm activity by attenuating the expression or activity of several virulence factors such as Shiga toxin [53].

In the current study, catechin attenuated the expression of the *acrA* gene related to *E. coli* biofilm formation and multidrug resistance. In a previous study, the expression of *acrA* was significantly reduced in multidrug-resistant *E. coli* strains under the pressure of four aminoglycosides (streptomycin, gentamicin, amikacin, and apramycin) at sub-inhibitory concentration [11]. Molecular docking was performed to achieve a better understanding of catechin action on *E. coli* biofilm-related genes. Based on the result, catechin exhibited high binding affinity to the AcrB protein, which is one of the AcrAB-TolC efflux pump proteins that are responsible for *E. coli* multidrug resistance [21]. The AcrAB-TolC efflux pump is an RND-type tripartite efflux pump that is a major contributor to multidrug resistance in Gram-negative bacteria. It has three major proteins, one of which is the AcrB protein, which represents the ligand interaction site. Upon ligand binding to AcrB, quaternary structural changes occurred in AcrB that communicated with AcrA to trigger structural changes leading to the opening of the TolC channel from the sealing resting state [54]. Although the AcrA protein lacks a substrate binding site, it serves as a link between AcrB and TolC and plays an important role in the stability of the AcrAB-TolC pump [54]. Based on the in silico result, we suggested that catechin isolated from *Canarium patentinervium* Miq. might indirectly reduce the expression of the biofilm related *acrA* gene by binding to the AcrB domain of the *E. coli* AcrAB-TolC efflux pump. Further in vitro tests to confirm the efflux pump inhibition effect of catechin are required.

## 4. Materials and Methods

### 4.1. Plant Material

The leaves and bark of *Canarium patentinervium* Miq. were previously collected from one individual tree in Bukit Putih, Selangor, Malaysia (3°5′24″ N 101°46′0″ E). The plant was collected with the approval and assistance of the local Indigenous people. The plant was identified by Mr. Kamaruddin (Forest Research Institute of Malaysia). A herbarium sample (PID 251210-12) has been deposited at the Forest Research Institute of Malaysia. The leaves and bark were air dried and ground into small particles using an industrial grinder. Our team previously isolated catechin from the ethanolic extracts of the leaves through bioassay-guided fractionation using Sephadex LH-20 (30 cm × 60 cm) and Silica gel (4 cm × 90 cm) [55].

### 4.2. Chemicals

The following chemicals were purchased from different manufacturers: MacConkey agar and Mannitol salt agar (Himedia, Thane, India), Nutrient agar, EMB agar medium, Mueller–Hinton agar medium and Mueller–Hinton broth medium (Oxoid, Hampshire, England), Brain Heart Infusion (BHI) broth and CCA medium (Condalab, Madrid, Spain), Gram stain solutions (Fluka, Buch, Switzerland), Glycerol (B.D.H, London, UK), Normal saline (Mediplast, Dubai, United Arab Emirates), Absolute ethanol 99% (Merck, Darmstadt, Germany), DMSO ≥ 99% (Merck, Darmstadt, Germany), EasyScript ^®^First-Strand cDNA Synthesis SuperMix (Transgen, Beijing, China), Real mod TM green (Takara, Maebashi, Japan), Quick-RNA Fungal/Bacterial Microprep Kit (Zymo, CA, USA).

### 4.3. Bacterial Strains

The clinical isolates of *E. coli* were obtained from the Bacteriology Unit, Department of Clinical Laboratory in the Medical City, Baghdad, Iraq. Samples were collected from UTI patients during the period from November 2021 to December 2021. *E. coli* ATCC reference bacteria (ATCC 25922, ATCC 8739, and ATCC 43895) were obtained from the Central Health Lab, Baghdad, Iraq. For all experiments except for the gene expression, three *E. coli* (ATCC reference) cultures and 15 clinical isolates were used.

Bacteria were cultured in a Brain Heart Infusion (BHI) broth medium and incubated at 37 °C for 24 h (Incubator IN55 Plus, Memmert GmBH, Schwabach, Germany) to promote bacterial growth. To identify *E. coli*, bacteria were streaked in general nutrient agar and differential culture media such as Chromogenic Coliforms Agar (CCA) medium and Eosin Methylene Blue agar (EMB). Antibiotic susceptibility (Table 5) was performed in VITEK^®^2 system (bioMérieux, Craponne, France) according to manufacturing company, *E. coli* was inoculated on MacConkey agar and incubated at 37 °C for 24 h.

### 4.4. Evaluation of Antibacterial Activity

#### 4.4.1. Disc Diffusion Assay

This test was performed using the Kirby–Bauer technique for disc diffusion following the National Committee for Clinical Laboratory Standards methods (NCCLS) [56]. Seven antimicrobial discs have been used (azithromycin 15 µg, vancomycin 30 µg, clindamycin 2 µg, erythromycin 10 µg, gentamicin 10 µg, rifampin 5 µg, and tetracycline 10 µg) (Bioanalyse, Ankara, Turkey) based on recommendations given by the Clinical Laboratory Standards Institute (CLSI,2020) [57]. The inoculum was prepared by transferring at least 3–5 isolated colonies that were grown previously in CCA agar to a bijou bottle containing 3 mL of normal saline and incubated at 35 °C for 2 h to achieve the turbidity of growth equal to the normal turbidity standard of 0.5 McFarland/625 nm (inoculation of 1 × 10^8^ CFU/mL).

The bacterial broth was used 15 min after the inoculum turbidity was adjusted. Then the inoculum was streaked into a petri dish with Mueller–Hinton agar medium with a thickness of 4 mm. The plate was dried at room temperature, then the antibiotic discs were added and incubated at 35 °C for 24 h. Catechin was used at a concentration of 100 mg/mL dissolved in DMSO ≥ 99%. The agar well diffusion method was performed for catechin based on the CLSI recommendation. A well with a diameter of 6 mm was punched aseptically into the petri dish with a sterile cork borer, and 20 µL of catechin solution was transferred into the well and incubated at 37 °C for 24 h [58]. Positive and negative controls were used, where the negative control included the solvent (DMSO ≥ 99%), and the positive control represented the antibiotic discs. The diameter of the zone of inhibition was measured using a digital vernier caliper to determine the microbial growth. The experiments were performed in triplicate and the mean values were presented.

#### 4.4.2. Minimum Inhibitory Concentration (MIC)

The broth microdilution method was performed based on the guidelines given by CLSI as described by Eloff [59]. A stock solution of catechin dissolved in DMSO ≥ 99% was prepared in 1.5 mL microcentrifuge tubes (Eppendorff) to a final concentration of 64 mg/mL and filtered with a 0.22 Millipore filter. Likewise, each individual antibiotic powder (Panpharma S.A., Luitré, France) was dissolved in a suitable solvent, DMSO ≥ 99% for (azithromycin, tetracycline, erythromycin, and rifampin) and distilled water for (gentamicin, vancomycin, and clindamycin) with an initial concentration of 64 mg/mL for all antibiotics. Two-fold serial dilutions were made from the stock solution to obtain a concentration range from 32 mg/mL to 0.125 mg/mL using Mueller–Hinton broth in 96-well plates.

A standardized bacterial suspension (100 μL) at a concentration of 1 × 10^6^ CFU/mL was added to each well containing the previously prepared 100 μL of diluted antimicrobial agents, resulting in a final volume of 200 μL in each well and final antibiotic concentrations ranging from 16 mg/mL to 0.125 mg/mL for a recommended final bacterial cell count of about 5 × 10^5^ CFU/mL. To find out how sensitive the bacteria being tested were, a positive control was made up of broth, the antimicrobial agent’s solvent (either distilled water or DMSO), and the bacteria. On the other hand, a broth without inoculum and an antimicrobial agent solvent served as the negative control. Then the microtiter plates were incubated at 37 °C for 24 h. After that, the MIC value was determined visually by recording the lowest concentration with no visible growth (the first clear well). MIC values were determined in triplicate and repeated to confirm activity.

#### 4.4.3. Minimum Bactericidal Concentration (MBC)

The minimum bactericidal concentration (MBC) was recorded as the lowest concentration that resulted in 99.9% killing of bacteria growth [60]. The MBC assay was performed using Ozturk and Ercisli method [61] by which only the susceptible bacteria from the MIC assay were considered. Ten microliters were taken from the well obtained from the MIC experiment (MIC value) and two wells above the MIC value well and spread on MHA plates. The number of colonies was counted after 18–24 h of incubation at 37 °C. The concentration of a sample that produces < 10 colonies is considered as MBC value. Each experiment was performed in triplicate and the MBC/MIC ratio was determined. If the ratio MBC/MIC ≤ 4, the effect is considered bactericidal, but if the ratio MBC/MIC > 4, the effect is defined as bacteriostatic [19,62].

#### 4.4.4. Synergistic Assay

In the presence of different antimicrobial agents (gentamicin, clindamycin, vancomycin, tetracycline, erythromycin, azithromycin, and Rifampin), the antimicrobial activity of catechin was tested. The bacterial strain was spread with a turbidity of 0.5 McFarland on Mueller–Hinton agar (MHA) plates. For the assessments of the synergistic effects, selected antibiotic discs were discretely impregnated with 5 μL of catechin (at the MBC value) and employed on the inoculated agar plates. Plates were incubated at 37 °C for 18 h. The zones of inhibition produced by catechin in combination with standard antibiotics after overnight incubation were estimated. If zones of combination treatment > ‘more than’ (zone of catechin + zone of the corresponding antibiotic), it was interpreted as synergism; if zone of combination treatment = (zone of catechin + zone of correspondence antibiotic), it was interpreted as additive; if zone of combination treatment < ‘less than’ (zone of catechin + zone of the corresponding antibiotic), it was interpreted as antagonism [63,64].

### 4.5. Evaluation of Antibiofilm Activity

#### 4.5.1. Biofilm Formation by *E. coli*

Microtiter plate assay or tissue culture plate (TCP) was used for quantitative determination of *E. coli* biofilm formation in accordance with O’Toole with some modification [65]. In brief, a sterile polystyrene tissue culture plate (composed of 96 flat bottom wells) was filled with 200 μL of the diluted prepared bacterial suspension and incubated at 37 °C for 24 h. Then, the content of each well was gently removed by tapping the plates, and the wells were washed twice with 200 μL phosphate buffer saline (PBS) (pH 7.2) to remove free-floating ‘planktonic’ bacteria. Biofilms formed by adherent ‘sessile’ bacteria on plate were fixed by placing them in the incubator at 37 °C for 30 min. Then, the wells were stained with 200 μL of 1% crystal violet solution for 45 min. After that, the microtiter plates were rinsed three times with sterile distilled water to remove excess dye and left to dry for 45 min at room temperature, then de-stained by adding 200 µL ethanol. A sterile bacterial broth was used as a negative control to identify non-specific binding. A micro-ELISA reader (at 570/655 nm wavelength) was used to measure the optical densities (OD) of stained bacterial biofilms. The experiment was conducted in triplicate and the average OD values were considered. The results were graded into strong, moderate, and non or weak biofilm.

#### 4.5.2. Biofilm Inhibition Assay

Catechin at MIC value was added to each well of a 96-well microplate except for the positive control well, which contained gentamicin and the negative control well, which contained bacterial broth only [66,67]. Bacterial culture (1 × 10^8^ CFU/mL) in an amount of 100 µL was pipetted into each well and the well was then incubated at 37 °C for 18 h. After that, the content of each well was removed, and the wells were rinsed three times with distilled water and allow to dry at 60 °C for 45 min. Then the wells were stained with 200 µL of 1% crystal violet and incubated at room temperature for 30 min. Finally, the plates were rinsed with distilled water, de-stained with ethanol, and incubated at room temperature for 15 min. A microplate ELISA reader (model 680, Bio-Rad, Hercules, CA, USA) at 570/655 nm was used to measure the optical densities. The experiment was conducted in triplicate and the mean absorbance was considered. The percentage of biofilm inhibition was then compared with the positive control and calculated according to the following formula:(1)$$\%\mathrm{Of}\mathrm{inhibition}=\frac{OD\;control-OD\;treatment*100}{OD\;control}$$

### 4.6. In Silico Study

#### Molecular Docking

A docking study was conducted to predict the target protein for catechin antibiofilm action. In a previous study, the *acrA* gene that is related to antibiotic resistance of *E. coli* biofilm was identified [11]. The crystal structure of the target protein was retrieved from the Protein Data Bank database (https://www.rcsb.org/, accessed on 25 April 2022). AcrA protein (PDB-ID: 5NG5) is a component of the *E. coli* AcrAB-TolC efflux pump and consists of an AcrA subunit with six chains and 373 sequence length, an AcrB subunit with three chains and 1049 sequence length, an AcrZ subunit with three chains and 55 sequence length, and a TolC protein with three chains and 493 sequence length. The target protein was prepared for docking by removing water, adding hydrogens and kollman charges and saved as pdb format using Autodock tools 1.5.1 [68]. Then the natural ligand was extracted from the protein using Discovery Studio Visualizer software (https://discover.3ds.com/discovery-studio-visualizer-download, accessed on 25 April 2022). The ligand and the protein were edited in Autodock tools 1.5.1. The grid box was centered on the ligand and control docking was run using Autodock Vina 1.1.2 software [69]. The same protocol was repeated with catechin as the ligand and the 3D structure of catechin (ID: 9064) was obtained from the PubChem data base (https://pubchem.ncbi.nlm.nih.gov/, accessed on 26 April 2022). The docking simulation was repeated three times, and the average binding affinity was considered. Several parameters were evaluated, such as the binding affinity of catechin toward the target protein, the ligand interaction site, the amino acids involved in the binding pocket, and the bonds formed at the interaction site. Results were analyzed using PyMOL [70] and visualized using Discovery Studio Visualizer (Dassault Systèmes, San Diego, CA, USA).

### 4.7. Gene Expression Using Quantitative Real-Time RT-PCR

To evaluate the effect of catechin on biofilm-related gene expression, quantitative real-time PCR (Mx3000P qPCR System, Agilent, Santa Clara, USA) was performed on *E.*
*coli* clinical isolates with and without catechin (at the sub-inhibitory concentration = 0.5 mg/mL) [71]. *E. coli* clinical isolates (2 × 10^5^ CFU) were inoculated in 250 mL of tryptic soy broth (TSB) [50]. Quick-RNA™ Fungal/Bacterial Microprep Kit was used to isolate the total RNA according to the manufacturer’s instructions (Zymo, USA). The RNA was treated with DNase/RNase-Free water to remove genomic DNA. The concentration and optical absorbance of each extracted RNA were confirmed with Nanodrop (Epoch Microplate Spectrophotometer, Agilent, Santa Clara, USA) at 260 nm/280 nm and samples were kept at −80 °C to elicit DNA contamination from total isolated RNA. EasyScript^®^ First-Strand cDNA Synthesis SuperMix kit was used to synthesize the cDNA according to the manufacturer’s instructions (Transgen/China). The cDNA reaction components including Random Primer (N9) in a volume of 1 µL, 2×ES Reaction Mix (10 µL), EasvScript^®^RT/RI Enzyme Mix (1 µL), RNase-free Water (Up to 20 µL), Eluted RNA (5 µL). qRT-PCR was performed by the SYBR Green gene expression assay [27]. In each sterile PCR tube, sample components of Real MODTM Green W2 2x qPCRmix in a volume of 10 µL, Forward Primer (10 µM) in a volume of 2.0 µL, Reverse Primer (10 µM) in a volume of 2.0 µL, Template DNA (4 µL), and DNase/RNase free water (up to 20 µL) were added. The thermal program was performed as follows: initial activation step, which was optimized at 95 °C for 10 min, 1 cycle; denaturation step, which was optimized at 95 °C, for 30 s; annealing was optimized at 60 °C, for 30 s; extension was optimized at 72 °C for 30 s. Denaturation, annealing, and extension steps were performed over 40 cycles. Then there was the final extension, which was optimized at 72 °C, for 5 min, 1 cycle. The 16S rRNA gene was used as a housekeeping gene, while the *acrA* gene was used as a target gene for *E. coli.* The primers used in this study are listed in Table 6.

### 4.8. Statistical Analysis

All experiments were performed in triplicate and data were expressed as mean ± standard deviation using GraphPad Prism 9 software for Mac, www.graphpad.com (GraphPad, San Diego, CA, USA). The data were analyzed using one-way ANOVA followed by Tukey test. For the gene expression, unpaired T-test was performed. A significant difference was considered at the level of *p* < 0.01. Data for the percentage of biofilm inhibiting activity of catechin were presented using Microsoft Excel.

## 5. Conclusions

Biofilm-forming *E. coli* places a significant burden on the health of the population. With a high rate of recurrence, *E. coli* becomes increasingly resistant to antimicrobial treatments and more difficult to eradicate. Catechin isolated from *Canarium patentinervium* Miq. exhibited strong biofilm inhibiting activity by reducing the expression of the biofilm-related *acrA* gene. This study highlights how important natural products are for treating infectious diseases that are resistant to the currently available antimicrobials.

## Figures and Tables

**Figure 1 antibiotics-11-01223-f001:**
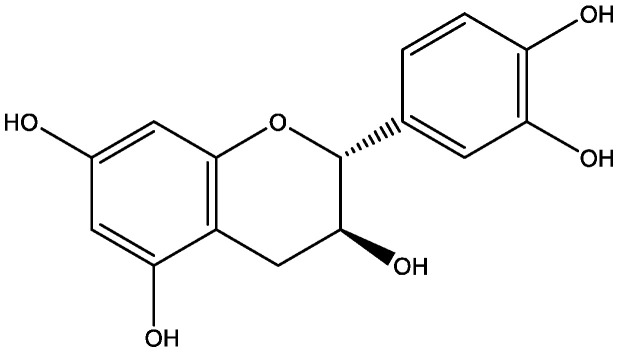
The chemical structure of catechin.

**Figure 2 antibiotics-11-01223-f002:**
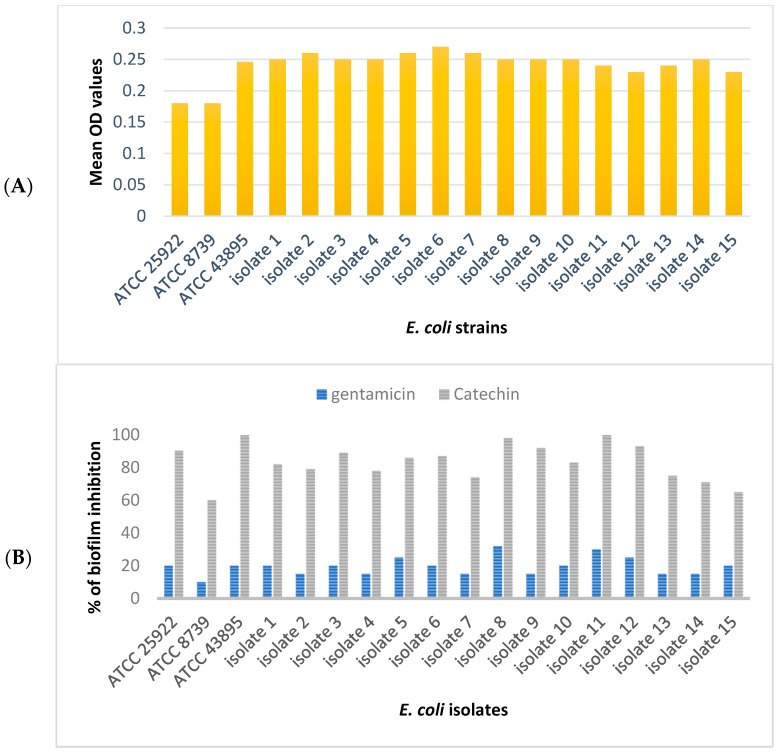
Biofilm formation ability of *E. coli* ATCC and clinical isolates (**A**), and biofilm inhibition effect of catechin (**B**). Gentamicin is the positive control. All the data were obtained from three independent experiments, and the mean values were presented.

**Figure 3 antibiotics-11-01223-f003:**
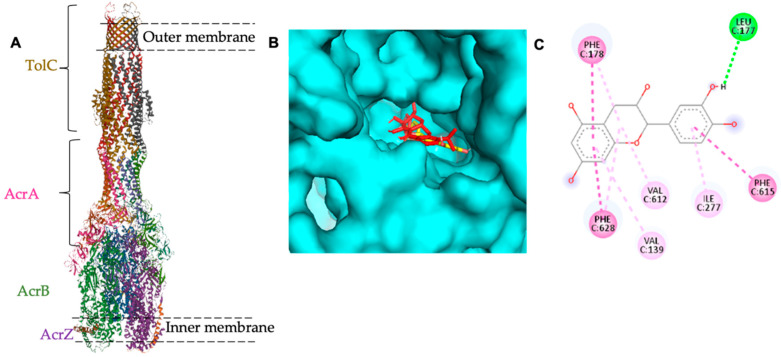
(**A**) Cryo-EM asymmetric structure of AcrBZ-Tolc pump of *E. coli* at 6.50 Å resolution (PDB-ID: 5NG5). (**B**) Binding affinity of catechin to AcrB protein of *E. coli* (docked catechin in yellow whereas docked minocycline is in red color). (**C**) the amino acids involved in the binding site of catechin to AcrB protein.

**Figure 4 antibiotics-11-01223-f004:**
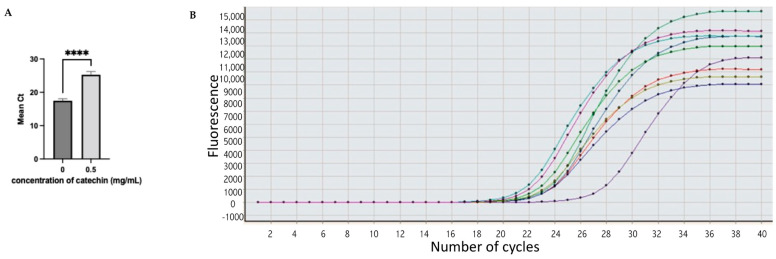
Expression of *acrA* gene in *E. coli* clinical isolates. (**A**) The level of expression without catechin treatment and with catechin treatment at concentration = 0.5 mg/mL. (**B**) The level of *acrA* expression in nine isolates represented in colored lines after catechin treatment. Data were expressed as mean ± SD with significant level **** *p* < 0.001.

**Table 1 antibiotics-11-01223-t001:** Zone of inhibition (mm) for the antimicrobial agents and catechin alone and in combination against *E. coli* isolates.

Bacteria	Antimicrobials		Zone of Inhibition (ZI) (mm)			
		ZI of Antimicrobials	ZI Breakpoints According to CLSI *	ZI of Catechin	ZI of Catechin in Combination with Antimicrobials	Outcome
***Escherichia coli*** **ATCC 25922 ^#^**	Rifampin	7.42 ± 0.0	R	10.47 ± 0.0	16 ± 0.0	Antagonism
	Tetracycline	-	R		10.4 ± 0.0	Additive
	Erythromycin	-	R		10.0 ± 0.0	Antagonism
	Clindamycin	-	R		10.45 ± 0.0	Additive
	Azithromycin	10.95 ± 0.0	R		22.5	Synergy
	Vancomycin	-	R		10.4 ± 0.0	Additive
	Gentamicin	31.32 ± 0.0	S		35 ± 0.0	Antagonism
***Escherichia coli*** **ATCC 8739 ^#^**	Rifampin	17.24 ± 0.0	I	9 ± 0.0	26 ± 0.0	Additive
	Tetracycline	-	R		9.1 ± 0.0	Additive
	Erythromycin	-	R		8.8 ± 0.0	Antagonism
	Clindamycin	-	R		9.0 ± 0.0	Additive
	Azithromycin	15.15 ± 0.0	S		25 ± 0.0	Synergy
	Vancomycin	-	R		9.5 ± 0.0	Synergy
	Gentamicin	13.14 ± 0.0	I		22 ± 0.0	Additive
***Escherichia coli*** **ATCC 43895 ^#^**	Rifampin	-	R	11.8 ± 0.0	11.5 ± 0.0	Antagonism
	Tetracycline	-	R		12.0 ± 0.0	Synergy
	Erythromycin	-	R		11.5 ± 0.0	Antagonism
	Clindamycin	-	R		11.45 ± 0.0	Antagonism
	Azithromycin	6.03 ± 0.0	R		17 ± 0.0	Synergy
	Vancomycin	-	R		11.8 ± 0.0	Additive
	Gentamycin	13.6 ± 0.0	I		22	Antagonism
** *Escherichia coli* ** **(isolate 1)**	Rifampin	2.45 ± 4.2	R	12.27 ± 1.3	13.89 ± 3.6	Antagonism
	Tetracycline	13.85 ± 2.5	I		28.58 ± 1.0	Synergy
	Erythromycin	11.97 ± 7.9	I		25.04 ± 0.6	Synergy
	Clindamycin	7.71 ± 8.7	R		22.19 ± 1.3	Synergy
	Azithromycin	11.58 ± 3.2	R		24.8 ± 0.5	Synergy
	Vancomycin	13.72 ± 7.2	I		26.43 ± 0.5	Synergy
	Gentamicin	10.42 ± 5.6	R		23.58 ± 0.4	Synergy
** *Escherichia coli* ** **(isolate 2)**	Rifampin	-	R	13.92 ± 3.6	14.0 ± 0.0	Additive
	Tetracycline	16.83 ± 1.0	S		31.5 ± 10.0	Synergy
	Erythromycin	18.97 ± 3.5	I		33.1 ± 0.0	Synergy
	Clindamycin	17.75 ± 0.4	I		31.5 ± 4.5	Additive
	Azithromycin	12.60 ± 2.84	I		27.2 ± 0.5	Synergy
	Vancomycin	17.87 ± 2.2	S		31.6 ± 0.9	Additive
	Gentamicin	16.95 ± 0.3	S		30.7 ± 0.8	Additive
** *Escherichia coli* ** **(isolate 3)**	Rifampin	-	R	12.9 ± 2.4	9.0 ± 5.2	Antagonism
	Tetracycline	12.35 ± 0.4	I		26.12 ± 0.2	Synergy
	Erythromycin	16.19 ± 1.0	I		31.5 ± 2.1	Synergy
	Clindamycin	15.37 ± 3.8	I		28.2 ± 3.1	Additive
	Azithromycin	12.43 ± 2.2	I		26.3 ± 0.3	Synergy
	Vancomycin	18.87 ± 0.0	S		32.0 ± 0.7	Synergy
	Gentamicin	16.24 ± 1.3	S		29.0 ± 0.1	Additive
** *Escherichia coli* ** **(isolate 4)**	Rifampin	10.87 ± 0.0	R	13.09 ± 0.0	20.21 ± 1.6	Antagonism
	Tetracycline	15.26 ± 0.5	S		30.0 ± 1.6	Synergy
	Erythromycin	20.61 ± 4.0	I		34.0 ± 0.0	Synergy
	Clindamycin	16.42 ± 1.1	I		29.5 ± 1.2	Additive
	Azithromycin	12.42 ± 0.7	I		26.1 ± 0.5	Synergy
	Vancomycin	20.03 ± 5.2	S		25.5 ± 0.1	Additive
	Gentamicin	-	R		13.1 ± 2.4	Additive
** *Escherichia coli* ** **(isolate 5)**	Rifampin	-	R	10.68 ± 5.6	10.71 ± 1.1	Additive
	Tetracycline	16.82 ± 2.3	S		30.23 ± 0.0	Synergy
	Erythromycin	16.5 ± 0.0	I		28.0 ± 4.4	Synergy
	Clindamycin	-	R		12.1 ± 3.3	Synergy
	Azithromycin	12.12 ± 1.5	I		24.0 ± 0.8	Synergy
	Vancomycin	17.71 ± 3.4	S		29.0 ± 0.3	Synergy
	Gentamicin	-	R		11.0 ± 0.2	Synergy
** *Escherichia coli* ** **(isolate 6)**	Rifampin	-	R	9.93 ± 3.6	9.0 ± 4.5	Additive
	Tetracycline	12.26 ± 1.0	I		25.0 ± 0.0	Synergy
	Erythromycin	14.1 ± 0.6	I		25.0 ± 0.0	Synergy
	Clindamycin	-	R		10.5 ± 0.0	Synergy
	Azithromycin	12.31 ± 2.6	I		23.2 ± 0.4	Synergy
	Vancomycin	16.82 ± 0.5	I		27.0 ± 0.5	Synergy
	Gentamicin	11.45 ± 0.0	R		21.8 ± 0.3	Synergy
** *Escherichia coli* ** **(isolate 7)**	Rifampin	-	R	12.75 ± 5.6	14.74 ± 8.3	Synergy
	Tetracycline	12.93 ± 0.2	I		27.3 ± 0.9	Synergy
	Erythromycin	14.59 ± 1.4	I		28.1 ± 3.4	Synergy
	Clindamycin	-	R		13.22 ± 0.2	Synergy
	Azithromycin	12.94 ± 0.6	I		26.6 ± 0.9	Synergy
	Vancomycin	18.21 ± 1.5	S		30.89 ± 0.6	Additive
	Gentamicin	11.81 ± 0.0	R		25.6 ± 0.4	Synergy
** *Escherichia coli* ** **(isolate 8)**	Rifampin	-	R	10.08 ± 4.3	14.69 ± 0.8	Synergy
	Tetracycline	12.12 ± 11.0	I		24.6 ± 5.3	Synergy
	Erythromycin	17.73 ± 8.76	I		28.4 ± 0.0	Synergy
	Clindamycin	-	R		11.54 ± 1.3	Synergy
	Azithromycin	12.04 ± 5.6	I		23.1 ± 0.2	Synergy
	Vancomycin	-	R		10.0 ± 3.4	Additive
	Gentamicin	11.02 ± 7.0	R		22.0 ± 1.1	Synergy
** *Escherichia coli* ** **(isolate 9)**	Rifampin	9.46 ± 1.1	R	13.58 ± 2.3	23.1 ± 2.5	Additive
	Tetracycline	13.46 ± 2.3	I		27.89 ± 6.8	Synergy
	Erythromycin	19.09 ± 0.6	I		33.1 ± 0.5	Synergy
	Clindamycin	-	R		14.0 ± 0.0	Synergy
	Azithromycin	12.82 ± 0.0	I		27.0 ± 0.4	Synergy
	Vancomycin	-	R		13.5 ± 0.6	Additive
	Gentamicin	11.1 ± 0.0	R		26.4 ± 0.8	Synergy
** *Escherichia coli* ** **(isolate 10)**	Rifampin	-	R	12.9 ± 1.7	12.6 ± 1.3	Additive
	Tetracycline	13.4 ± 0.5	I		27.0 ± 1.1	Synergy
	Erythromycin	14.34 ± 4.1	I		28.5 ± 8.85	Synergy
	Clindamycin	15.38 ± 3.2	I		30.0 ± 5.5	Synergy
	Azithromycin	12.55 ± 8.5	I		26.4 ± 0.5	Synergy
	Vancomycin	-	R		12.82 ± 0.9	Additive
	Gentamicin	13.52 ± 6.1	I		27.2 ± 0.4	Synergy
** *Escherichia coli* ** **(isolate 11)**	Rifampin	-	R	10.38 ± 0.6	10.4 ± 0.5	Additive
	Tetracycline	12.28 ± 0.0	I		24.5 ± 0.7	Synergy
	Erythromycin	-	R		10.5 ± 1.4	Additive
	Clindamycin	16.69 ± 1.1	I		28.1 ± 1.1	Synergy
	Azithromycin	12.21 ± 0.9	I		22.5 ± 0.5	Additive
	Vancomycin	16.59 ± 3.1	I		29.0 ± 0.7	Synergy
	Gentamicin	13.76 ± 2.5	I		25.4 ± 1.1	Synergy
** *Escherichia coli* ** **(isolate 12)**	Rifampin	8.76 ± 5.3	R	13.17 ± 1.5	22.5 ± 6.4	Synergy
	Tetracycline	18.34 ± 7.1	I		31.9 ± 0.5	Synergy
	Erythromycin	-	R		13.4 ± 11.0	Additive
	Clindamycin	15.71 ± 2.6	I		30.5 ± 0.0	Synergy
	Azithromycin	12.77 ± 1.5	I		26.4 ± 0.2	Synergy
	Vancomycin	15.24 ± 1.1	I		28.5 ± 0.4	Additive
	Gentamicin	13.73 ± 0.6	I		27.5 ± 0.9	Synergy
** *Escherichia coli* ** **(isolate 13)**	Rifampin	-	R	13.18 ± 8.7	13.0 ± 0.0	Additive
	Tetracycline	9.82 ± 4.3	R		25.4 ± 1.4	Synergy
	Erythromycin	-	R		14.21 ± 0.0	Synergy
	Clindamycin	18.27 ± 5.2	I		32.0 ± 0.5	Synergy
	Azithromycin	12.67 ± 9.7	I		27.1 ± 0.0	Synergy
	Vancomycin	16.37 ± 10.0	I		29.45 ± 0.5	Additive
	Gentamicin	13.14 ± 8.7	I		27.2 ± 0.3	Synergy
** *Escherichia coli* ** **(isolate 14)**	Rifampin	7.67 ± 2.6	R	12.4 ± 1.3	20.0 ± 3.8	Additive
	Tetracycline	10.22 ± 11.0	R		23.4 ± 1.5	Synergy
	Erythromycin	-	R		12.42 ± 0.5	Additive
	Clindamycin	-	R		13.1 ± 0.6	Synergy
	Azithromycin	11.28 ± 5.4	I		23.5 ± 2.7	Additive
	Vancomycin	16.24 ± 8.6	I		29.0 ± 0.4	Synergy
	Gentamicin	11.81 ± 5.9	I		25.1 ± 0.2	Synergy
** *Escherichia coli* ** **(isolate 15)**	Rifampin	-	R	12.62 ± 6.6	12.5 ± 7.8	Additive
	Tetracycline	15.89 ± 7.4	S		29.1 ± 0.0	Synergy
	Erythromycin	9.42 ± 11.0	R		24.01 ± 11.6	Synergy
	Clindamycin	-	R		14.0 ± 4.3	Synergy
	Azithromycin	-	R		12.5 ± 0.9	Additive
	Vancomycin	15.37 ± 8.3	I		28.0 ± 0.9	Additive
	Gentamicin	-	R		13.2 ± 0.6	Synergy

^“#”^ means statistical data are unavailable, “-” means no activity, and “*” indicates that the results are categorized according to Clinical and Laboratory Standards Institute (CLSI) guidelines into R resistant; I intermediately resistant; and S sensitive. All the data were obtained from three independent experiments and is expressed as mean ± SD.

**Table 2 antibiotics-11-01223-t002:** Minimum bactericidal concentration (MBC) (mg/mL), minimum inhibitory concentration (MIC) (mg/mL), and MBC/MIC ratio for the antimicrobial agents and catechin against *E. coli* isolates.

Bacteria	Effect	Rifampin	Tetracycline	Erythromycin	Clindamycin	Azithromycin	Vancomycin	Gentamycin	Catechin
***Escherichia coli*** **ATCC 25922 ^#^**	MBC	8	32	32	8	8	64	4	2
	MIC	4	16	8	4	4	16	2	1
	MBC/MIC	2(+)	2(+)	4(+)	2(+)	2(+)	4(+)	2(+)	2(+)
***Escherichia coli*** **ATCC 8739 ^#^**	MBC	4	32	32	16	1	32	4	4
	MIC	2	16	16	8	0.5	16	2	2
	MBC/MIC	2(+)	2(+)	2(+)	2(+)	2(+)	2(+)	2(+)	2(+)
***Escherichia coli*** **ATCC 43895 ^#^**	MBC	8	32	16	8	16	16	4	2
	MIC	4	16	8	4	8	8	2	1
	MBC/MIC	2(+)	2(+)	2(+)	2(+)	2(+)	2(+)	2(+)	2(+)
** *Escherichia coli* ** **(isolate 1)**	MBC	32	32	32	32	2	32	8	4
	MIC	16	8	4	4	0.5	8	4	1
	MBC/MIC	2(+)	4(+)	8(−)	8(−)	4(+)	4(+)	2(+)	4(+)
** *Escherichia coli* ** **(isolate 2)**	MBC	16	32	16	32	2	32	8	4
	MIC	8	4	4	4	1	8	4	1
	MBC/MIC	2(+)	8(−)	4(+)	8(−)	2(+)	4(+)	2(+)	4(+)
** *Escherichia coli* ** **(isolate 3)**	MBC	16	32	32	32	1	16	8	4
	MIC	8	8	4	4	0.5	2	4	1
	MBC/MIC	2(+)	4(+)	8(−)	8(−)	2(+)	8(−)	2(+)	4(+)
** *Escherichia coli* ** **(isolate 4)**	MBC	32	64	32	32	1	32	8	4
	MIC	16	4	8	8	0.5	8	4	1
	MBC/MIC	2(+)	16(−)	4(+)	4(+)	2(+)	4(+)	2(+)	4(+)
** *Escherichia coli* ** **(isolate 5)**	MBC	16	64	32	32	1	32	8	4
	MIC	8	4	8	2	0.5	8	4	1
	MBC/MIC	2(+)	16(−)	4(+)	16(−)	2(+)	4(+)	2(+)	4(+)
** *Escherichia coli* ** **(isolate 6)**	MBC	16	32	16	32	2	64	4	4
	MIC	8	4	4	8	1	16	1	1
	MBC/MIC	2(+)	8(−)	4(+)	4(+)	2(+)	4(+)	4(+)	4(+)
** *Escherichia coli* ** **(isolate 7)**	MBC	32	32	64	32	2	64	16	4
	MIC	8	8	8	2	1	8	8	1
	MBC/MIC	4(+)	4(+)	8(−)	16(−)	2(+)	8(−)	2(+)	4(+)
** *Escherichia coli* ** **(isolate 8)**	MBC	64	32	32	32	4	32	32	4
	MIC	8	8	4	4	1	16	16	1
	MBC/MIC	8(−)	4(+)	8(−)	8(−)	4(+)	2(+)	2(+)	4(+)
** *Escherichia coli* ** **(isolate 9)**	MBC	64	32	32	32	2	32	4	2
	MIC	16	8	8	4	1	8	2	0.5
	MBC/MIC	4(+)	4(+)	4(−)	8(−)	2(+)	4(+)	2(+)	4(+)
** *Escherichia coli* ** **(isolate 10)**	MBC	64	32	32	32	4	64	4	2
	MIC	32	8	4	4	1	16	2	0.5
	MBC/MIC	2(+)	4(+)	8(−)	8(−)	4(+)	4(+)	2(+)	4(+)
** *Escherichia coli* ** **(isolate 11)**	MBC	64	32	16	32	2	16	8	4
	MIC	32	8	4	4	0.5	4	2	1
	MBC/MIC	2(+)	4(+)	4(+)	8(−)	4(+)	4(+)	4(+)	4(+)
** *Escherichia coli* ** **(isolate 12)**	MBC	64	32	64	32	4	32	8	4
	MIC	32	8	8	4	1	8	2	1
	MBC/MIC	2(+)	4(+)	8(−)	8(−)	4(+)	4(+)	4(+)	4(+)
** *Escherichia coli* ** **(isolate 13)**	MBC	64	32	16	16	2	32	8	4
	MIC	32	4	8	4	0.5	8	4	1
	MBC/MIC	2(+)	8(−)	2(+)	4(+)	4(+)	4(+)	2(+)	4(+)
** *Escherichia coli* ** **(isolate 14)**	MBC	16	64	32	32	2	32	8	4
	MIC	8	32	4	4	1	4	4	1
	MBC/MIC	2(+)	2(+)	8(−)	8(−)	2(+)	8(−)	2(+)	4(+)
** *Escherichia coli* ** **(isolate 15)**	MBC	32	64	32	32	1	32	8	4
	MIC	16	32	4	4	0.5	16	4	1
	MBC/MIC	2(+)	2(+)	8(−)	8(−)	2(+)	2(+)	2(+)	4(+)

^“#”^ means statistical data are unavailable. All the data were obtained from three independent experiments. For the MBC/MIC ratio, (+) bactericidal; (−) bacteriostatic.

**Table 3 antibiotics-11-01223-t003:** Binding affinity of catechin to AcrB protein of *E. coli.*

Compound	Molecular Docking Binding Affinity ΔG (Kcal/mol)	Residue Involved in the Binding Site	Bonds Involved in the Binding Site
Minocycline (control)	−8.8	PHE-178 (chain C), ASN-274 (chain C)	Hydrogen bond
		VAL-612 (chain C), ALA-279 (chain C)	Pi-Pi stacked
		ILE-277 (chain C), PHE-615 (chain C)	Pi-Sigma
Catechin	−8.2	GLY-179 (chain C)	Hydrogen bond
		LEU-177 (chain C)	Carbon Hydrogen bond
		PHE-178 (chain C), VAL-612 (chain C), ILE-277 (chain C)	Pi-Pi stacked

**Table 4 antibiotics-11-01223-t004:** The expression level of the 16S rRNA and *acrA* genes in *E. coli* clinical isolates.

Isolate Number	Mean of Ct of 16S rRNA (Untreated)	Mean of Ct of 16S rRNA (Treated)	Mean of Ct of *acrA* Gene (Untreated)	Mean of Ct of *acrA* Gene (Catechin Treated)	ΔCt for *acrA* Gene	Result
**1**	17	19.34	17.8	24.8	7	Down
**2**	16.9	18.45	17.2	24.7	7.5	Down
**3**	17.2	19.5	-	-	-	-
**4**	17.2	18.45	-	-	-	
**5**	17.3	21.5	16.8	24.7	7.9	Down
**6**	17.6	19.34	18.3	25	6.9	Down
**7**	18.2	19	-	-	-	-
**8**	17.7	21	-	-	-	-
**9**	18	20.41	17.8	24.8	7	Down
**10**	18.1	19.45	16.6	26.7	10.1	Down
**11**	17.6	18	-	-		-
**12**	17.7	18.2	18	27.2	9.2	Down
**13**	16.9	17.7	17.7	24.8	7.1	Down
**14**	16.7	17.9	16.9	24.7	7.8	Down
**15**	17.3	19.3	-	-	-	-

Ct, thermal cycle; -, acrA gene is not detected. ∆Ct = Ct_acrA_ (treated with catechin) − Ct_acrA_ (untreated).

**Table 5 antibiotics-11-01223-t005:** Bacterial source and antibiotic resistance profile.

Bacteria	Source	Resistance Profile
*Escherichia coli* ATCC 25922	Central Health Lab/Iraq	(R) TET, ERY, VAN, RIF, AZM, PIP (S) GEN, MIN, MEM
*Escherichia coli* ATCC 8739	Central Health Lab/Iraq	(R) TET, ERY, CLI, VAN (S) GEN, AZM, RIF
*Escherichia coli* ATCC 43895	Central Health Lab/Iraq	(R) RIF, TET, ERY, AZM, PIP, TIC, TOB (S) CFM, FEP, MIN
*Escherichia coli*	Urine from UTI samples	(R) TIC, PIP, GEN, TOB, CIP, SXT (S) TIM, TZP, CAZ, FEP, ATM, IPM, MEM, AMK, MIN

(R) resistant, (S) sensitive, TET tetracycline, ERY erythromycin, VAN vancomycin, RIF rifampin, AZM azithromycin, PIP piperacillin, GEN gentamycin, MIN minocycline, MEM meropenem, CLI clindamycin, TIC ticarcillin, TOB tobramycin, CFM cefixime, FEP cefepime, CIP ciprofloxacin, SXT trimethoprim-sulfamethoxazole, TIM ticarcillin- Clavulanic acid, TZP piperacillin-tazobactam, CAZ ceftazidime, ATM aztreonam, IPM imipenem, AMK amikacin.

**Table 6 antibiotics-11-01223-t006:** The primer sequences for the real time qPCR analysis.

Genes	Type	Sequences (5′–3′)	Temperature (C)
**16 sRNA (reference gene)**	Forward Reverse	AGAGTTTGATCMTGGCTCAG CTGCTGCSYCCCGTAG	50 52
** *acrA* ** **gene (target gene)**	Forward Reverse	TTGAAATTCAGGAT CTTAGCCCTAACAGGATGTG	53 57.2

## Data Availability

The data used and/or analyzed in this study are available from the corresponding author on reasonable request.

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
