# Peer review of "Evaluation of Catechin Synergistic and Antibacterial Efficacy on Biofilm Formation and *acrA* Gene Expression of Uropathogenic *E. coli* Clinical Isolates"

_antibiotics, 2022, doi:10.3390/antibiotics11091223_

Round 1

Reviewer 1 Report

This paper reports the antibacterial and antibiofilm activities, and gene expression of catechin isolated from Canarium patentinervium Miq. against multidrug resistant E. coli. The aims of this paper are interesting and the experimental analysis was apparently well conducted. However, there are several points that need explanation and/or correction. Please see the comments below.

1. The title must be revised in order to better reflect the work. Still in the title: Canarium patentinervium: please, use the italic form. Also, the same should be observed for In vitro and In silico (please, review the entire manuscript);

2. Abstract: abbreviations should be defined the first time they appear such as MIC and MBC;

3. The keywords should be concise, specific and relevant. Terms appearing in the title should be avoided. Use new terms that make it possible to broaden the search for this work in the databases;

 4- Introduction:

-  Reference numbers should be placed in square brackets [ ]. Please, review this throughout the manuscript;

- Lines 35 to 38: there is a linguist problem with this sentence. The logic and the structure are flawed; it needs to be corrected;

- Line 53: “ESBL” and “AmpC” - abbreviations should be defined the first time they appear – please rephrase;

- Line 55: the same applies to: RND-based tripartite efflux pump- AcrAB-TolC – please define it;

- Lines 63-65: this sentence is confused – please rephrase;

- Lines 74-76: there is a linguist problem with this sentence. The logic and the structure are flawed; it needs to be corrected;

 5- Results section:

- Line 81: “The current study investigates ...” – please rewrite to: “The study investigated...”

- Line 88: Figure 2 - ATCC 43895 – please remove italic form;

- Line 94: please, define MIC - according to the instructions for the authors, abbreviations should be defined the first time they appear in each section;

- Line 95: “... against two of the tested strains (E. coli ATCC 8739 and the clinical E. coli isolates)” – according to abstract and methodology section, the authors tested fifteen clinical isolates; so there is an inconsistency in this sentence – please rephrase;

- Lines 95-97: “While the remaining antiniotics...” - there is a linguist problem with this sentence. The logic and the structure are flawed; it needs to be corrected;

- Line 103: please, define MBC - according to the instructions for the authors, abbreviations should be defined the first time they appear in each section;

- Line 106: “showed cidal activity” – what it means? – please rephrase;

- Line 108: “While catechin had bactericidal activity against all the tested E. coli strains.” - there is a linguist problem with this sentence. The logic and the structure are flawed; it needs to be corrected;

-  Line 117 – Table 1:

a) the table should be presented right after being cited in the text;

b) the table appears to be unconfigured; in the format presented it was impossible to understand the results - the table and the caption should be improved;

c) according to the caption “15 clinical isolates of E. coli.” are presented; however, the table only shows the result of a single isolate – please explain this better;

- Line 130: Figure 1:

a) please, remove the title of the figure;

b) please improve the figure: also show the isolated effect of catechin and each antibiotic, in addition to the synergistic effect;

c) what does this plate represent? what antibiotics were tested? which strain or isolate is shown? Please, remove this figure;

- Lines 154-158: please, transfer this sentence to the discussion;

- Lines 168-169: “While gentamycin...” - there is a linguist problem with this sentence. The logic and the structure are flawed; it needs to be corrected;

- Figure 2:

a) please, remove the titles of the figures A and B (tranfer it to the caption);

b) what does the image next to figure A mean? What the authors intend to show is not clear in this image - please remove it;

c) please, improve the legend of the figure B - put only gentamicin and catechin;

d) the authors should clarify why they represent the result found for 15 clinical isolates as unique;

 6- Discussion section

- Line 334: please rewrite in vitro in italic form;

- Lines 336-338: “In addition…” - there is a linguist problem with this sentence. The logic and the structure are flawed; it needs to be corrected;

 7- Material and Methods section

- Line 394: Please, start the sentence with: The following chemicals were purchased from different manufacturers: …

- Line 394-400: please, provide in parentheses (Manufacturer, City, Country) for all the chemicals;

- From line 401: please, provide all device names in the following format: device (Model, Company, City, Country);

- Lines 438, 442, 448, and 451: Please, provide the concentration of DMSO used;

- Line 456: the sentence must not start with a number. I suggest: “A standardized bacterial suspension (100 µL) at a concentration...”;

- Line 492: please, rewrite E. coli in italic form;

 8- Conclusion section: the conclusion presents a summary of the results. It should finalize the findings presented and point out perspectives for the advancement of knowledge in the area studied – please review;

9- References - please, review the formatting according to the guidelines described in the instructions for authors;

In my final comments, I recommend that the manuscript should be widely reviewed by the authors. The introduction, material and methods, results and discussion, and conclusion sections must be reformed in order to explain more concisely the effects of catechin isolated from Canarium patentinervium against E. coli strains.

The manuscript must undergo a comprehensive English review as well as a grammar review.

Reviewer 2 Report

In this work Jubair et al investigate the antimicrobial and anti-biofilm properties of a natural product (catechin) against various reference and clinical strains of Escherichia coli. The authors demonstrate that catechin has inherent antimicrobial properties as well as synergistic effects with some antibiotics. They further show that catechin interacts with the RND efflux pump AcrAB-TolC subunit AcrB, and that catechin treatment reduces expression of acrA. Overall, this study identifies interesting antimicrobial properties of catechin and suggests a possible mechanism through modulation of efflux. However, there are several limitations related to data presentation, analysis, and interpretation that need to be addressed prior to acceptance. Additionally, the manuscript contains several typos, grammatical errors, and unclear sentences throughout need to be corrected prior to publication.

Major issues:

- table 1 is difficult to interpret as is. Please move labelling to the side to clarify what each row means. Also, please indicate in the table how these zones of inhibition and MIC would be categorized clinically based on CLSI guidelines rather than just providing the mm zone size.

- the paper claims to have used 15 clinical isolates, but most figures have one “E. coli” class for clinical isolates. Did they only test one clinical isolate or were all clinical isolates averaged together? It would be more appropriate to show all isolates individually as seen in table 3.

- figure 2 is lacking statistical analysis and error bars

- does catechin have synergistic effects with antibiotics against biofilms? Would catechin be effective against preformed biofilms?

- lines 215-216: unclear how these proteins were identified and why they are relevant to this study based on the text

- lines 225-232: AcrB is not the molecular target of the antibiotics listed. AcrB interacts with these proteins to pump them out of the cell

- the manuscript suggests that catechin has antimicrobial effects by interacting with AcrB. Does catechin actually inhibit efflux or does it interact with AcrB solely because it gets pumped out of cells through ArcAB-TolC?

- figure 5 and Table 3: Ct values must all be normalized to the abundance of housekeeper and not presented as raw Ct values to avoid bias based on RNA abundance. Also, in table 3 is the mean Ct of 16S rRNA for the treated samples, untreated samples, or both together? The fold changes cannot be interpreted unless the housekeeper Ct for both samples is presented individually or unless the target gene abundance is normalized to housekeeper abundance in the same sample. See Pfaffl NAR 2001 for more information.

- unclear how catechin binding to AcrB would have such a profound effect on acrA transcription seeing as AcrB is neither a transcription factor nor signaling protein. Please expand on the proposed mechanism.

- if catechin exerts antimicrobial effects through AcrAB, these effects should be eliminated in an acrAB deficient strain. This should be tested to determine whether the activity of catechin is solely AcrAB dependent or if other pathways are involved.

Minor:

- the title implies the paper will investigate the expression of catechin, but instead it investigates the effects of catechin on expression of other genes. This could be clarified.

- numerous typos, grammatical errors, and unclear phrasing throughout

- please conform to standard protein and gene nomenclature for bacteria

- would be nice to have a structure of catechin presented in the paper

- line 36: E. coli pathotypes are not capitalized (e.g. uropathogenic and enterotoxigenic E. coli)

- lines 39-40: adhesins not adhesions

- line 42: CAUTI means catheter-associated UTI, not community acquired

- line 62: “ancient”

- throughout: should be spelled gentamicin not gentamycin

- figure 1: include catechin alone control

- lines 162-164: crystal violet absorbance values are highly variable day to day and between isolates, so it makes little sense to bin weak vs strong biofilm formation on raw absorbance values alone without any sort of normalization

- figure 3c: please make the residue labels bigger and easier to read

- it would be interesting to know how conserved AcrB and particularly the highlighted residues are across UPEC and E. coli more broadly

- should be 16S rRNA not 16 sRNA

- figure 5b: please provide labeling to indicate what these curves represent

- lines 308-312: very unclear what is meant here. Also unclear how changes in Ct relate to resistance vs sensitivity in this context

Round 2

Reviewer 1 Report

The authors performed an extensive review of the manuscript and all suggestions and corrections were addressed.

However, in conclusion the terms E. coli (line 532) and Canarium patentinervium (line 534) must be written in italic form.

Reviewer 2 Report

This paper has been significantly improved in clarity of language and data presentation, and the authors have addressed my comments. 

- still need to italicize gene names "acrA"

- in the conclusion E. coli and Canarium patentinervium are not italicized

- in line 95 clarify what "best synergistic combination" means - most consistent? strongest synergistic effect?
